# Excessive manganese content in children's multivitamin supplements: Potential for neurodevelopmental harm and other adverse health outcomes

Seth H. Frisbie[1]*, Erika J. Mitchell[2], Amy Hoeltge[3], Lindsey A. Pett[4], Molly G. Hoeltge[5]

1 Department of Chemistry and Biochemistry, Norwich University, Northfield, Vermont, United States of America, 2 Better Life Laboratories, Incorporated, East Calais, Vermont, United States of America, 3 Department of General and Organic Chemistry, University of Vermont, Burlington, Vermont, United States of America, 4 Department of Earth and Environmental Sciences, Norwich University, Northfield, Vermont, United States of America, 5 Lake Champlain Waldorf School, Shelburne, Vermont, United States of America

* sfrisbie@norwich.edu

## Abstract

### Background

Manganese (Mn) is an essential nutrient vital for many physiological processes, but it can cause adverse health effects at high exposures, particularly during neurodevelopment. Mn is included in many children's multivitamins and mineral supplements, but it has not been documented whether the levels of Mn in such supplements are appropriate or safe. In this study, we measured Mn concentrations in children's multivitamin and mineral supplements and compared the concentrations to the Institute of Medicine's (IOM's) tolerable Upper intake Levels (ULs).

### Methods

We purchased 52 multivitamin and mineral supplements in various forms, including tablets, capsules, gummies, powders, and liquids. Of these samples, half listed supplemental Mn on their labels; half did not. We quantified their manganese concentrations using inductively coupled plasma-optical emission spectroscopy. We compared the measured Mn concentrations of the samples to the labeled concentrations and to the IOM ULs.

### Results

Of the 26 products with supplemental Mn on their labels, 23 (88.5%) contained more Mn than shown on the label, while 3 (11.5%) contained less. The mean percent difference between labeled and measured Mn content was 42.5%. One product (4%), a liquid, would exceed the IOM's Mn ULs based on labeled

**Data availability statement:** All files are available from the Harvard Dataverse repository (https://doi.org/10.7910/DVN/16XWHT).

**Funding:** The author(s) received no specific funding for this work.

**Competing interests:** The authors have declared that no competing interests exist. EJM's affiliation is with Better Life Laboratories, Inc., a nonprofit organization that conducts scientific research and provides technical expertise, equipment, and training to help needy people around the world. Better Life Laboratories received no specific funding for this project from any donors. Donors to Better Life Laboratories provided no input in choosing the subject matter of this project, the hypotheses that were tested, the method of analysis, the research findings, or the manner of disseminating the results. This does not alter our adherence to PLOS ONE policies on sharing data and materials.

concentrations alone if consumed daily, 5 products (19%) would exceed IOM ULs based on measured concentrations, and 9 (34.6%) would exceed IOM ULs based on measured concentrations if used as supplements to ordinary diets. Of the 26 products without supplemental Mn listed on their labels, 13 (50.0%) contained measurable Mn concentrations. None of the products without supplemental Mn listed on their labels would exceed IOM ULs, even if consumed as supplements to ordinary diets.

## Conclusions

Some children's multivitamin supplements may lead to exceedances of IOM ULs for Mn, especially when consumed as supplements to ordinary diets.

---

## Introduction

Approximately one-third of children under 19 years of age in the United States (US) use multivitamin supplements, according to National Health and Nutrition Examination Survey (NHANES) data from 2017 to 2018 [1]. Nearly 20% of the best-selling multivitamin supplements for children under 5 years available from a major US online vendor list manganese (Mn) as an ingredient on their labels (S1 Table). Recent research has associated high Mn exposure with various neurodevelopmental effects in children, while other studies have shown that low Mn exposure can also be potentially of concern [2]. Given that hundreds of thousands of children are exposed to Mn through daily multivitamin supplements, it is vital to ensure that the amounts of Mn contained in these products are appropriate for healthy development.

In adults, excess Mn has long been recognized as a neurotoxin, causing manganism, which presents with balance issues, tremors, and mental and mood changes similar to Parkinson's Disease [3]. Perhaps because Parkinson's Disease is considered to be mainly a disease of older adults, it was not until recently that studies of excess exposures to Mn even included young adults or children as subjects [4].

However, once children's neurodevelopment in the context of Mn exposures began to be investigated, researchers noted numerous neurodevelopmental problems associated with high Mn exposures, including behavioral problems, IQ deficits, attention deficits, and memory problems [2]. Significantly, U-shaped curves for Mn exposures were reported in some studies, with both low and high exposures associated with worse outcomes than moderate exposures [2]. Such U-shaped curves would be expected, given that Mn, while toxic at high doses, is also an essential nutrient, a critical component of many enzymes, and it also plays a role in bone formation and immune function.

It is also possible that excess Mn may be associated with other types of problems besides neurological harm, as shown by recent cross-sectional studies using data from large population surveys of health and nutrition, such as NHANES in the United States. Notably, these cross-sectional studies are observational in nature

and merely show associations between Mn biomarkers and health conditions. Nevertheless, they suggest possible associations that may merit further research. These recent cross-sectional studies have noted associations between high blood, serum, or urine Mn and anemia [5,6], decreased bone density [7–9], cardiovascular disease [10], kidney disease [11,12], liver disease [13–18], and sarcopenia [19].

While most of these studies examining associations between Mn biomarkers and health conditions included only adults, some studies examined adolescents and children. Cross-sectional studies focusing on children or adolescents noted associations between Mn biomarkers and anemia [20], decreased bone density [21], blood pressure and kidney parameters [22], hepatic steatosis [23], lung function [24], and dental caries [25]. While it is not known whether these observed associations are indicative of causal effects between Mn biomarkers and health effects in children, the associations suggest areas for further research.

Unfortunately, many of these cross-sectional studies did not investigate the possibility of adverse health outcomes at both ends (high and low) of exposure. Nevertheless, in the studies that did compare the extremes of exposures to more moderate exposures, U-shaped curves were commonly reported, including for kidney disease [11], anemia [6,20], cognitive performance [26], insulin resistance [27], diabetic retinopathy [28], kidney disease [11], liver disease [13], all-cause mortality [29], sarcopenia [30], and obesity [31]. In these studies U-shaped curves describing the associations between Mn exposure and adverse health outcomes identified inversion points from 7.32 to 13.45 µg/L of Mn in blood [5,6,13,28–30,32].

Unfortunately, direct associations between dietary Mn intake and biomarker concentration in blood, serum, and urine cannot be drawn from NHANES data because the NHANES food intake data do not include Mn content [33]. Other similar national surveys, such as the United Kingdom's National Diet and Nutrition Survey (NDNS) and the Chinese Health and Nutrition Survey (CHNS), typically measure either Mn biomarkers or estimates of dietary intakes, but not both [34,35]. Nevertheless, in a smaller study that measured both dietary Mn intake and biomarkers of exposure, direct associations were reported between high dietary Mn intake and low serum ferritin levels; however, the specific daily Mn intake amounts associated with low serum ferritin levels were not identified [36].

Mn deficiency in humans eating normal diets is quite rare, and it is quite easy to obtain the required amounts of Mn in any diet that includes food intake [37], so Mn supplementation is likely unnecessary for most people. On the other hand, the risks of excessive Mn intakes are becoming better known as they are more thoroughly investigated. These risks may include neurodevelopmental harm in children, as well as other possible health concerns such as anemia or reduced bone density, as suggested by the recent cross-sectional studies discussed above [2,20].

Given the adverse health outcomes associated with high Mn exposure, we believe it is important to directly measure and quantify the amounts of Mn found in children's multivitamin supplements. Although there are currently no data linking excess Mn exposure from dietary supplements to direct harm to children, the data from observational and cross-sectional studies suggest a potential for harm. In this study, we used inductively coupled plasma-optical emission spectroscopy (ICP-OES) to measure the actual amount of Mn found in popular children's multivitamin supplements. We then compared these amounts to the labeled values. The accuracy of these measured Mn concentrations was evaluated by comparison with results from a standard reference material and by known additions of standard to samples. Precision was determined by analyzing duplicate samples. We also calculated daily Mn intakes from supplements. We compared them with Adequate Intakes (AIs) and tolerable Upper intake Levels (ULs) published by the Institute of Medicine (IOM) [38]. We hypothesized that the amounts of Mn contained in the children's supplements would be accurately reflected on the labels and that daily Mn intakes from these supplements would help meet sufficiency requirements while not exceeding the tolerable levels defined by the IOM. We also hypothesized that the cost/dose of the supplements would be associated with the quality of labeling; that is, more expensive supplements would have labels that more accurately represented the Mn content, and that more expensive supplements would be less likely to exceed IOM tolerable levels.

 

## Materials and methods

### Sample selection

We identified Amazon.com as the leading vendor of dietary supplements in the US market, with nationwide availability and annual sales that outpace all competitors, both online and in brick-and-mortar stores [39]. A search using the term "children's multivitamin with Mn" yielded 152 results at www.amazon.com on December 18, 2021. All 26 available products from these search results with "infants", "babies", "kids", "toddlers", or "children" and a Mn salt listed on their label were purchased. These 26 samples are assumed to be a reasonable representation of children's multivitamins with supplemental Mn available in the United States (US).

This same search also yielded 48 products from 34 brands with "infants", "babies", "kids", "toddlers", or "children" and no Mn salt listed on their label. A random number generator was used to select 26 available products from 26 different brands. This second set of 26 samples was purchased and assumed to be a reasonable representation of children's multivitamins without supplemental Mn sold in the US.

Thus, a total of 52 products were purchased from different brands in the US in January, 2022. Twenty-six of the 52 products listed a Mn salt on their labels. Of these 26 products, 3 were labeled "capsule", 8 were labeled "chewable", 5 were labeled "gummy", 3 were labeled "liquid", 3 were labeled "powder", and 4 were labeled "tablet." Twenty-six of these 52 products did not list a Mn salt on their labels. Of these 26 products, 2 were labeled "chewable", 20 were labeled "gummy", 3 were labeled "liquid", and 1 was labeled "tablet." Although the forms of the products differ somewhat between those listing Mn on the label and those not listing Mn on the label, we do not assume that the likelihood of inclusion of Mn and other minerals is independent of the form of the supplements; gummies may be less likely to include minerals [S1 Table]. Since the focus of this study is on the labeled and actual Mn content of the supplements, not the forms of these supplements, we did not consider these observed differences in forms to be significant for the scope of our study. Of the 52 products we purchased, 50 listed 3 or more vitamins and minerals on the label, while 2 listed only 1 vitamin or mineral on the label. Samples were stored in their original packaging at 18–24° C until resources could be secured for chemical analyses in June to August of 2025.

To assess the accuracy of our results, we also purchased Multielement Tablets Standard Reference Material® (SRM 3294) from the National Institute of Standards and Technology (NIST).

### Sample preparation

Solutions of all 52 samples and the Multielement Tablets Standard Reference Material® (SRM 3294) tablets were prepared for chemical digestion using laboratory-grade glassware. A microwave instrument (CEM Discover Explorer Hybrid 12®) with closed microwave vessels was employed for sample digestion. Approximately 0.200 to 0.230 grams (g) of each homogenized sample was weighed out in the reaction vessel and mixed with 10.00 milliliters (mL) of deionized water, 2.00 mL of concentrated nitric acid ($HNO_3$; Fisher Chemical Lot 207552), and 0.50 mL of 30% hydrogen peroxide ($H_2O_2$; Fisher Chemical Lot 222582). The masses of these samples have 3 significant figures; therefore, most of the results in this paper are reported to the same precision. The vessels were reacted for approximately 5 minutes, then sealed and placed in the autosampler of the microwave instrument. The vessels were then heated to 120° Celsius (C) and maintained at that temperature for 30.0 minutes. The digestates were cooled to room temperature and made to 25.00 mL with deionized water. If needed, these digestates were subsequently diluted with additional deionized water to achieve a measured Mn concentration within the range used for calibration.

### Sample analysis

Mn concentrations were quantified using ICP-OES with a SPECTRO ARCOS instrument (SPECTRO Analytical Instruments, Kleve, Germany). Each sample was introduced into the plasma as an aerosol via a nebulizer, where it was

 

atomized and excited within an argon plasma operating at approximately 10,000 Kelvin (K). The SPECTRO ARCOS detected emitted light at element-specific wavelengths using an axial viewing configuration to optimize sensitivity and accuracy [40,41].

Standard solutions at 0.000, 5.006, 9.994, 14.997, and 19.983 mg/L of total Mn were prepared from an ISO 17034 and ISO 17025 certified stock solution (IV-28–125ML; Inorganic Ventures Lot T2-MEB724442) and were used by the instrument software for calibration. Samples were analyzed for Mn using a SPECTRO ARCOS ICP-OES, according to the manufacturer's recommended operating conditions. The instrument was operated at a plasma power of 1,400 watts (W) with a peristaltic pump speed of 30 revolutions per minute (rpm). Argon gas flows were maintained at 13.50 liters per minute (L min⁻¹) for the coolant, 1.20 L min⁻¹ for the auxiliary flow, and 0.75 L min⁻¹ for the nebulizer. Mn concentrations were determined using the Mn emission line at 257.611 nanometers (nm), which was selected based on manufacturer guidance for high sensitivity and minimal spectral interference [41]. The manufacturer's proprietary software determined the calibration equation [41]. All standards and samples were analyzed in triplicate, and the mean concentrations were reported and used for statistical analysis. This work was done at Norwich University in Northfield, VT.

### Quality assurance and quality control

Quality assurance and quality control measures included the analysis of the Multielement Tablets Standard Reference Material® (SRM 3294) from the NIST in triplicate. In addition, after every analytical window of 13 samples, we also measured a reagent blank, a duplicate sample, and a known addition of standard to a sample (also called "sample matrix spikes" or "laboratory-fortified samples"). An additional reagent blank was analyzed immediately after calibration. Therefore, there were a total of 5 reagent blanks, 4 duplicate samples, and 4 known additions of standard to a sample during the analysis of all 52 samples. The analysis of the Standard Reference Material and the known additions of standard to samples were used to evaluate accuracy. The analysis of the duplicate samples was used to evaluate precision [42].

### Calculations and statistical analysis

Differences between measured and labeled concentrations were calculated by subtracting the labeled values from the measured concentrations. Percent differences between measured and labeled concentrations were calculated by dividing the absolute value of these differences by the average of the measured and labeled values, then multiplying by 100. Thus, percent differences are always positive. Percent differences were calculated only for the 26 products with supplemental Mn.

Associations between cost/dose, labeling accuracy, and appropriateness of dosage with respect to IOM dietary reference values were explored using statistical tests. Statistical analyses were performed in R version 4.5.1, "Great Square Root." The normality of the data was evaluated prior to any hypothesis testing using the Shapiro-Wilk test. Non-parametric methods (Wilcoxon rank sum test, Spearman's rank order correlation) were used for hypothesis testing for data with non-normal distributions. A significance level of $p < .05$ was assumed for all statistical tests.

## Results and discussion

### The detection limit, accuracy, and precision of the determination of Mn by inductively coupled plasma optical emission spectroscopy

The detection limit is the smallest concentration that can be reported with reasonable certainty for a given analytical procedure [43]. There are many ways to calculate a detection limit [43]. For this study, we defined our detection limit as the upper limit of a 1-tailed 99% confidence interval from the 5 separately prepared 0.00 mg/L Mn standards that were analyzed as samples. This approach is called the detection limit based on control charts [44]. This detection limit was 0.001 mg/L for digestates analyzed by ICP-OES, which equals 0.0001 mg/g for a sample of supplement product when 0.200 g of this sample is digested and diluted to 25.00 mL. These detection limits are rounded to one significant figure.

The measured concentrations of Mn in 3 analyses of the Multielement Tablets Standard Reference Material (SRM 3294) were 1.36 milligrams per gram of tablet (mg/g), 1.43 mg/g, and 1.44 mg/g by inductively coupled plasma-optical emission spectroscopy. All 3 concentrations fall within the acceptable range of 1.33 mg/g to 1.55 mg/g established by the NIST [45]. Therefore, the measured Mn concentrations for all 52 samples analyzed by inductively coupled plasma-optical emission spectroscopy are considered accurate according to this first definition of accuracy [42].

The percent recoveries of known additions of standard to samples were measured at 92.7%, 99.0%, 102%, and 112% by ICP-OES. The 95% confidence interval for these recoveries ranges from 88.7% to 114%. Since this range includes 100%, the measured Mn concentrations for all 52 samples analyzed by ICP-OES are considered accurate according to this second definition of accuracy [42]. In other words, these results suggest there are no statistically significant spectral interferences or matrix effects.

The precision of samples, based on the analysis of 4 samples in duplicate, is 0.007 mg/g. This precision is excellent, given that it does not vary until the thousandths of a mg/g. In contrast, the Multielement Tablets Standard Reference Material (SRM 3294) varies at the hundredths of a mg/g [42,45].

In conclusion, the percent recoveries of the Multielement Tablets Standard Reference Material (SRM 3294) by ICP-OES are highly accurate. The known additions of standard to samples measured by this method are highly accurate. The precision of samples by this method is highly precise.

Table 1 compares the sample preparation methods and analytical techniques used in this study and other similar prior studies. Our study used methods similar to those of comparable studies and achieved one of the lowest detection limits.

**Table 1. Comparison of preparation methods and analytical techniques to previous studies in similar matrices worldwide.**

| Country | Matrix | Sample Preparation | Analytical Technique | Detection Limit |
|---|---|---|---|---|
| United States (present study) | children's multivitamin/multimineral supplements | acidic oxidation with microwave digestion | ICP-OES | 0.001 mg/L = 1 µg/L<br>0.0001 mg/g = 0.1 µg/g |
| Brazil [46] | multivitamin formulations | *In vitro* gastrointestinal digestion (INFOGEST protocol) 2.0 | ICP-OES | 0.,1 µg/g [sic] |
| Turkey [47] | dietary supplements and dietary products | acidic oxidation with microwave digestion | ICP-OES | 2.5 µg/L |
| Czech Republic [48] | multivitamin preparations and dietary supplements | acidic oxidation with microwave digestion | ICP-OES | 2.1 µg/L |
| Serbia [49] | herbal food supplements | acidic oxidation with microwave digestion | ICP-MS | not stated |
| United States [50] | multivitamin/multimineral supplements | wet-ashing | ICP-MS | not stated |
| Nigeria [51] | dietary supplements | acidic oxidation with microwave digestion | ICP-MS | 0.019 µg/L |
| Pakistan [52] | multivitamins | acidic oxidation and heating | FAAS | not stated |
| Romania [53] | pharmaceutical products for veterinary use | ashing followed by acidic oxidation with heating | FAAS | not stated |
| Brazil [54] | Multivitamin/multimineral supplements | acidic oxidation with microwave digestion | FAAS | 0.021 mg/g |
| Brazil [55] | Food supplements for weight loss | drying | WDXRF | 20 (LOQ; units not stated) |

Abbreviations: ICP-OES = inductively coupled plasma atomic emission spectroscopy; ICP-MS = inductively coupled plasma mass spectrometry; FAAS = flame atomic absorption spectrometry; GFAAS = graphite furnace atomic absorption; WDXRF = wavelength-dispersive X-ray spectroscopy; LOQ = limit of quantification.

## The Mn concentrations reported on labels compared to the concentrations measured in samples

Labeled and measured concentrations of Mn for the 52 children's supplement products are shown in Table 2.

As shown in Table 2, 23 of the 26 (88.5%) products with a Mn salt listed on their labels contained more Mn than was shown on the label, while 3 (11.5%) products contained less. One product (sample 19) appeared to have a typographical error on the label; the Mn content was claimed to be "1 mcg" (interpreted as 1 microgram, µg) per serving when "1 mg" (1 milligram, mg) was most likely intended. This likely typographical error resulted in a 199 percent difference between the measured and labeled Mn concentrations for this sample (Table 2). The mean percent difference between the measured and labeled concentrations for the remaining 25 products with a Mn salt listed on their labels was 36.2%. The mean percent differences between labeled and measured Mn (42.5% for all samples or 36.2% excluding the sample with the presumed typographical error) were considerably higher than the 24% mean percent difference between labeled and measured Mn content reported by the United States Department of Agriculture survey of children's multivitamin products published in 2017 [50].

Notably, 13 of the 26 (50.0%) products without supplemental Mn listed on their labels contained measurable Mn concentrations. The mean measured Mn concentration in the products with no Mn salt listed on their labels was 0.0059 mg/g, while the mean measured Mn concentration in the products with a Mn salt listed on their labels was 0.501 mg/g; this difference was statistically significant ($W(52) = 10$, $p < .001$).

The cost of a one-day supply of product for a 5-year-old ranged from $0.07 to $1.70. There was no difference in average cost between products with a Mn salt listed on their labels ($0.51) and those with no Mn salt listed on their labels ($0.48) ($W(52) = 319$, $p = .735$). The cost of a one-day supply was also not correlated with the magnitude of difference between the labeled and measured Mn content ($r_s(50) = -0.13$, $p = .368$).

## Regulations applicable to Mn in children's supplements in the US

Dietary supplements, including children's multivitamins, are regulated in the US by the Dietary Supplement Health and Education Act of 1994 (DSHEA) [56]. This act states that the government should not "impose unreasonable regulatory barriers limiting or slowing the flow of safe products to consumers." To avoid imposing regulatory barriers, the DSHEA classifies dietary supplements as foods; thus, they are not subject to the special regulations that apply to drugs and medications.

Regarding safety, the government reserves the right to deem supplements adulterated that present "significant or unreasonable risks of illness or injury" under "conditions of use recommended or suggested in labelling" [56]. However, the burden of proof of adulteration, according to such a definition, is explicitly placed with the government. Thus, there are no specific regulations regarding Mn content in children's supplements in the US. Manufacturers are forbidden from selling unsafe products. However, if a product turns out to be unsafe, the government must provide specific evidence of the risks and harms before it can deem the product adulterated and remove it from the market.

The National Institutes of Health (NIH) in the US relies on the recommendations of the Food and Nutrition Board (FNB) of the National Academies of Sciences, Engineering, and Medicine (NASEM), a congressionally chartered non-profit institution, for its Daily Reference Intakes (DRIs). The U.S. FDA uses the DRIs developed by the FNB for its mandatory nutrition label regulations [57]. In the case of Mn, the FNB DRIs include Adequate Intakes (AIs) and tolerable Upper intake Levels (ULs) [58,59]. The IOM originally developed the FNB DRIs, so they are often referred to in the literature as the IOM DRIs [38]. The IOM DRIs for Mn for infants and children aged 8 years and younger are shown in Table 3 [38].

These IOM AIs and ULs were drawn from a review of nutritional studies published in 2001 [38]; thus, they do not reflect any of the vast research on Mn sufficiency and toxicity published over the last 25 years. The IOM did not find data specific to nutritional adequacy for Mn, so they based the AIs on the average (for 0–12 months) or median (for 1–8 years) recorded intakes of Mn in food [38]. More specifically, regarding children 1–8 years, they stated, "There are insufficient data to set an Estimated Average Requirement (EAR) for manganese for children ages 1 through 3 years. Therefore,

 

**Table 2. Labeled and measured concentrations of Mn in the 52 children's supplement products in milligrams of Mn (Mn) per gram of product (mg/g), the percent differences between labeled and measured Mn content, and the costs per day for a 5-year-old. Samples with measured concentrations below the nominal 0.0001 mg/g detection limit are labeled as ND (not detected). All concentrations are reported to 3 significant figures or to the nearest 0.0001 of a mg/g.**

| Sample | Labeled Mn (mg/g) | Measured Mn (mg/g) | Difference between Labeled and Measured Mn (mg/g) | % Difference between Labeled and Measured Mn | Cost per Day for 5-Year-Old |
|---|---|---|---|---|---|
| 1 | 1.79 | 2.52 | 0.729 | 33.9% | $0.34 |
| 2 | 1.15 | 0.737 | −0.410 | 43.5% | $0.41 |
| 3 | 0.708 | 0.832 | 0.124 | 16.2% | $0.41 |
| 4 | 0.740 | 0.831 | 0.0910 | 11.6% | $0.17 |
| 5 | 0.696 | 0.894 | 0.198 | 24.8% | $0.20 |
| 6 | 0.804 | 0.904 | 0.100 | 11.7% | $0.77 |
| 7 | 0.565 | 0.656 | 0.0914 | 15.0% | $0.65 |
| 8 | 0.611 | 0.561 | −0.0499 | 8.51% | $0.50 |
| 9 | 0.167 | 0.192 | 0.0256 | 14.3% | $0.32 |
| 10 | 0.133 | 0.299 | 0.166 | 76.8% | $0.43 |
| 11 | 0.158 | 0.217 | 0.0593 | 31.6% | $0.14 |
| 12 | 0.138 | 0.172 | 0.0346 | 22.3% | $0.99 |
| 13 | 0.115 | 0.245 | 0.130 | 72.4% | $0.31 |
| 14 | 0.0303 | 0.0473 | 0.0170 | 43.8% | $0.42 |
| 15 | 0.0041 | 0.0064 | 0.0023 | 43.7% | $0.43 |
| 16 | 0.0041 | 0.0067 | 0.0026 | 49.1% | $0.38 |
| 17 | 0.136 | 0.158 | 0.0213 | 14.5% | $1.69 |
| 18 | 0.0029 | 0.0071 | 0.0042 | 84.6% | $0.43 |
| 19 | 0.0001 | 0.0371 | 0.0370 | 199% | $0.70 |
| 20 | 0.144 | 0.376 | 0.230 | 88.8% | $0.96 |
| 21 | 0.0532 | 0.0939 | 0.0407 | 55.4% | $0.50 |
| 22 | 0.125 | 0.0957 | −0.0293 | 26.5% | $0.89 |
| 23 | 0.753 | 0.856 | 0.103 | 12.8% | $0.28 |
| 24 | 0.378 | 0.724 | 0.346 | 62.8% | $0.37 |
| 25 | 0.556 | 0.692 | 0.136 | 21.7% | $0.14 |
| 26 | 0.726 | 0.879 | 0.153 | 19.1% | $0.42 |
| 27 | | 0.0280 | | | $0.07 |
| 28 | | ND | | | $0.56 |
| 29 | | ND | | | $0.67 |
| 30 | | 0.0002 | | | $0.31 |
| 31 | | 0.0004 | | | $0.73 |
| 32 | | ND | | | $0.74 |
| 33 | | 0.108 | | | $0.32 |
| 34 | | ND | | | $0.14 |
| 35 | | 0.005 | | | $0.87 |
| 36 | | ND | | | $0.53 |
| 37 | | 0.0002 | | | $0.13 |
| 38 | | 0.0002 | | | $0.33 |
| 39 | | ND | | | $0.44 |
| 40 | | ND | | | $0.30 |
| 41 | | 0.0063 | | | $0.68 |
| 42 | | ND | | | $0.13 |

*(Continued)*

| Sample | Labeled Mn (mg/g) | Measured Mn (mg/g) | Difference between Labeled and Measured Mn (mg/g) | % Difference between Labeled and Measured Mn | Cost per Day for 5-Year-Old |
|---|---|---|---|---|---|
| 43 | | 0.0002 | | | $0.13 |
| 44 | | 0.0002 | | | $0.27 |
| 45 | | ND | | | $0.50 |
| 46 | | ND | | | $0.47 |
| 47 | | 0.0024 | | | $0.49 |
| 48 | | 0.0005 | | | $0.50 |
| 49 | | ND | | | $0.28 |
| 50 | | ND | | | $1.70 |
| 51 | | 0.0007 | | | $0.35 |
| 52 | | ND | | | $0.87 |
| Mean for Mn labeled products | 0.411 s=0.433 | 0.501 s=0.531 | 0.0905, s=0.184 | 42.5%, s=0.402% | $0.51, s=$0.34 |
| Mean for products without labeled Mn | | 0.0117, s=0.0299 | | | $0.48, s=$0.34 |
| Mean for Mn labeled products, omitting sample #19 | | | | 36.2%, s=0.248% | |
| Minimum for Mn labeled products | 0.0001 | 0.0064 | −0.410 | 8.51% | $0.14 |
| Maximum for Mn labeled Products | 1.79 | 2.52 | 0.729 | 199% | $1.69 |
| Minimum for products without labeled Mn | | ND | | | $0.07 |
| Maximum for products without labeled Mn | | 0.108 | | | $1.70 |

**Table 3.** The Institute of Medicine's (IOM's) Dietary Reference Intakes (DRIs) for manganese (Mn) in milligrams per day (mg/day) for infants and children aged 8 years and younger [58,59].

| Age | Adequate Intakes (AIs) | Tolerable Upper intake Levels (ULs) |
|---|---|---|
| 0-6 months | 0.003 mg/day | No Data |
| 7-12 months | 0.6 mg/day | No Data |
| 1-3 years | 1.2 mg/day | 2 mg/day |
| 4-8 years | 1.5 mg/day | 3 mg/day |

median intake data were used to set the AI. Data from the Food and Drug Administration Total Diet Study indicate a median intake of 1.22 mg/day of manganese for children aged 1 through 3 years" [38]. For "*Ages 4 through 13 years.* There have been a few manganese balance studies with children and all are subject to the caveats previously discussed. Therefore, they were not considered in setting an EAR. The Total Diet Study indicates a median intake of 1.48 mg/day for children aged 4 through 8 years" [38]. That is, the average intakes for infants or median intakes for children were redefined as the Adequate Intakes (AIs), so individuals not consuming these average or median amounts were classified as undernourished with regard to Mn. This decision would logically tend to increase the median Mn intake across populations actively seeking "sufficiency."

For its ULs in children over 1 year old, the IOM scaled down its adult UL of 11 mg/day by relative body weight and applied the result to children. For infants less than 1 year old, the scaling approach was not used due to "concern about

the infant's ability to handle excess amounts." The IOM goes on to remark, "To prevent high levels of Mn intake, the only source of intake for infants should be from food or formula" [38]. It must be kept in mind that the adult UL of 11 mg/day was also derived as the maximum intake from a dietary study that did not measure any health outcomes; a total of approximately 2 adult women consumed this maximum intake of 11 mg/day during a 3-day dietary survey, while the remaining 98 women in the study consumed lower amounts of Mn [38,60].

Mn metabolism is quite complex, with body levels of Mn controlled by homeostatic mechanisms [61]. Relatively high intakes of Mn from foods or beverages, such as tea, are not directly correlated with blood or serum Mn levels [62]. In contrast, Mn intakes through drinking water or Mn supplements have been correlated with elevated blood, serum, or hair Mn levels, suggesting that the Mn from supplements may be more readily absorbed than regular dietary Mn consumed in a complex matrix of other nutrients [63–66].

In sum, while the U.S. FDA does not regulate maximum Mn content in foods, it relies on the recommendations of the FNB of the NASEM, for the values used in its mandatory labeling regulations [57]. Thus, these DRI values are probably the most appropriate reference values for determining whether US products contain appropriate amounts of nutrients. The NASEM was formerly known as the IOM, and the DRI values for Mn and other mineral nutrients have not been changed since they were last revised by the IOM, so they are commonly referred to as the IOM DRI values.

The IOM AIs used in the US are based not on actual sufficiency, but rather on averages and medians of dietary intakes, so they should more accurately be termed "average intakes" rather than "adequate intakes." Intakes near these levels that are below the stated AIs may not actually be inadequate, but might more accurately be termed "below average intakes." More data are needed to determine the nutritional adequacy of Mn, particularly data identifying blood Mn levels that protect against adverse health outcomes and dietary patterns that might lead to these levels.

In contrast, the IOM ULs are based on maximum adult dietary intakes from a single study without associated health data [38,60]. Extremely high blood Mn levels have been associated with a variety of adverse health outcomes. If there are any predictive relationships between dietary Mn and blood Mn, then the extreme dietary Mn intakes from the single study used to define the ULs may not be entirely without risk [60]. Certainly, if children, who are actively undergoing neurological and physical development, were to proportionately exceed these maximum recorded adult dietary intakes by consuming supplemental Mn, this would potentially be of concern.

## Daily exposure estimates for Mn

**Daily exposure estimates based on labeled values.** Daily exposure estimates based on labeled Mn content for each age for which the samples are labeled for use are shown in Table 4.

Of the 26 products with a Mn salt listed on their labels, 1 out of these 26 (3.85%) products exceeded an IOM UL. This product was labeled for use by infants 6 months and older and provides 0.8 mg of Mn per day for infants 6–12 months old. For infants below 1 year old, the IOM notes, "To prevent high levels of Mn intake, the only source of intake for infants should be from food or formula" [38]; thus, any supplement containing detectable amounts of Mn exceeds the IOM recommendation for this age.

An additional 3 out of 26 (11.5%) products have labeled Mn content that equals the IOM UL for specified ages (Tables 3 and 4). While these labeled values do not exceed the IOM ULs, they leave no room for any further Mn exposures through either food or water. Consumption of any food or water that contains Mn in conjunction with a daily dose of one of these supplements will lead to exceedances of the IOM ULs, based on the labeled content of the supplements.

**Daily exposure estimates based on measured values.** Daily exposure estimates based on measured Mn content for each age for which the samples are labeled for use are shown in Table 5.

Of the 26 products with a Mn salt listed on their labels, 4 (15.4%) exceeded the IOM UL for 1- to 3-year-olds (Tables 3 and 5). Notably, 3 of these 26 (11.5%) products are identical to those whose labeled content equaled the IOM ULs; see Table 4. An additional product provides enough Mn to exceed the IOM UL for 4–5-year-olds (Tables 3 and 5). This product

**Table 4. Daily exposure estimates based on the labeled manganese (Mn) content at different ages, ranging from 6 months to 5 years. Only samples with labeled Mn content are included. Separate estimates are provided for each age covered by the indications on the label.**

| Sample | Daily Intakes of Mn Based on Labeled Values* | | | | | |
|---|---|---|---|---|---|---|
| | 6 Months (mg/day) | 1 Year (mg/day) | 2 Years (mg/day) | 3 Years (mg/day) | 4 Years (mg/day) | 5 Years (mg/day) |
| *1* | | | *2* | *2* | 2 | 2 |
| 2 | | | | | 1 | 1 |
| 3 | | 0.375 | 0.375 | 0.375 | 0.75 | 0.75 |
| *4* | | | *2* | *2* | 2 | 2 |
| 5 | | | 1.15 | 1.15 | 2.3 | 2.3 |
| 6 | | | | | 2 | 2 |
| 7 | | | 1 | 1 | 2 | 2 |
| 8 | | | | | 1 | 1 |
| 9 | | | | 0.5 | 0.5 | 0.5 |
| 10 | | 0.3 | 0.3 | 0.3 | 0.3 | 0.3 |
| 11 | | | | | 0.25 | 0.25 |
| 12 | | | | | 0.9 | 0.9 |
| 13 | | | | | 0.6 | 0.6 |
| 14 | | | | | 0.1 | 0.1 |
| 15 | | | 0.01 | 0.01 | 0.02 | 0.02 |
| 16 | | | | 0.01 | 0.02 | 0.02 |
| **17** | **0.8** | 1.5 | 1.5 | 1.5 | 2.3 | 2.3 |
| 18 | | 0.05 | 0.05 | 0.05 | 0.05 | 0.05 |
| 19 | | | 0.001 | 0.001 | 0.001 | 0.001 |
| 20 | | | | | 1.5 | 1.5 |
| 21 | | | | | 0.5 | 0.5 |
| 22 | | | 0.25 | 0.25 | 0.25 | 0.25 |
| *23* | | | *2* | *2* | 2 | 2 |
| 24 | | | | | 1 | 1 |
| 25 | | | 0.5 | 0.5 | 1 | 1 |
| 26 | | | | | | 0.4 |

* Labeled values which exceed the IOM ULs for corresponding age classes are highlighted in **bold**. Labeled values which equal the IOM ULs are highlighted with ***bold italics***.

(sample 20) contained more than twice its labeled value. None of the products with no Mn salt listed on their labels exceeded the IOM ULs.

**Daily exposure estimates based on ordinary dietary intakes in addition to measured values in supplements.** Children's vitamin and mineral products are intended to be consumed as supplements to ordinary intakes from food and beverages. Unfortunately, NHANES does not include Mn intakes in its What We Eat in America (WWEIA) surveys [33], so there are currently no nationwide representative surveys of dietary Mn intakes. Nevertheless, as noted above, the IOM defined its AIs for Mn as equal to the average recorded intakes for infants younger than 12 months, or the median recorded intakes of Mn from food for children older than 12 months from earlier surveys [38]. For 7-month-old infants, the IOM stated the average Mn intake was 500 µg/day, while for 12-month-old infants, the IOM stated the average Mn intake was 720 µg/day, which the IOM combined to 600 µg/day for the entire 7- to 12-month-old age class [38]. For children 1–3 years old, the IOM stated the median daily Mn intake as 1.22 mg/day, and for children 4–8 years old, the IOM

**Table 5. Daily exposure estimates in milligrams per day (mg/day) based on measured manganese (Mn) content at different ages from 6 months to 5 years. Separate estimates are provided for each age covered by the indications on the label.**

| Sample | Daily Intakes of Mn (Mn) Based on Measured Values* | | | | | |
|---|---|---|---|---|---|---|
| | 6 Months | 1 Year | 2 Years | 3 Years | 4 Years | 5 Years |
| | (mg/day) | (mg/day) | (mg/day) | (mg/day) | (mg/day) | (mg/day) |
| **1** | | | **2.82** | **2.82** | 2.82 | 2.82 |
| 2 | | | | | 0.643 | 0.643 |
| 3 | | 0.441 | 0.441 | 0.441 | 0.882 | 0.882 |
| **4** | | | **2.25** | **2.25** | 2.25 | 2.25 |
| 5 | | | 1.48 | 1.48 | 2.95 | 2.95 |
| 6 | | | | | 2.25 | 2.25 |
| 7 | | | 1.16 | 1.16 | 2.32 | 2.32 |
| 8 | | | | | 0.918 | 0.918 |
| 9 | | | | 0.577 | 0.577 | 0.577 |
| 10 | | 0.674 | 0.674 | 0.674 | 0.674 | 0.674 |
| 11 | | | | | 0.344 | 0.344 |
| 12 | | | | | 1.13 | 1.13 |
| 13 | | | | | 1.28 | 1.28 |
| 14 | | | | | 0.156 | 0.156 |
| 15 | | | 0.0156 | 0.0156 | 0.0312 | 0.0312 |
| 16 | | | | 0.0165 | 0.0330 | 0.0330 |
| **17** | **0.925** | 1.85 | 1.85 | 1.85 | 2.78 | 2.78 |
| 18 | | 0.123 | 0.123 | 0.123 | 0.123 | 0.123 |
| 19 | | | 0.610 | 0.610 | 0.610 | 0.610 |
| **20** | | | | | **3.10** | **3.10** |
| 21 | | | | | 0.950 | 0.950 |
| 22 | | | 0.205 | 0.205 | 0.205 | 0.205 |
| **23** | | | **2.27** | **2.27** | 2.27 | 2.27 |
| 24 | | | | | 1.92 | 1.92 |
| 25 | | | 0.622 | 0.622 | 1.24 | 1.24 |
| 26 | | | | | | 0.484 |
| 27 | | | 0.0191 | 0.0191 | 0.0382 | 0.0382 |
| 30 | | 0.0012 | 0.0012 | 0.0012 | 0.0012 | 0.0012 |
| 31 | | | 0.0009 | 0.0009 | 0.0017 | 0.0017 |
| 33 | | | 0.324 | 0.324 | 0.648 | 0.648 |
| 35 | | | 0.0117 | 0.0117 | 0.0235 | 0.0235 |
| 37 | | | 0.0005 | 0.0005 | 0.0010 | 0.0010 |
| 38 | | | 0.0013 | 0.0013 | 0.0013 | 0.0013 |
| 41 | | | | | 0.0250 | 0.0250 |
| 43 | | | 0.0004 | 0.0004 | 0.0009 | 0.0009 |
| 44 | | | 0.0007 | 0.0007 | 0.0015 | 0.0015 |
| 47 | | | 0.0101 | 0.0101 | 0.0202 | 0.0202 |
| 48 | | | 0.00436 | 0.00436 | 0.00436 | 0.00436 |
| 51 | | | 0.0018 | 0.0018 | 0.0054 | 0.0054 |

* Products that exceed the IOM ULs for corresponding age classes based on measured Mn content are highlighted in **bold**. Samples whose measured Mn content was below the detection limit are omitted.

stated the median daily Mn intake as 1.48 mg/day [38]. Table 6 presents daily Mn exposure estimates when the products that we tested are consumed as supplements to food in ordinary diets, where intakes from food are assumed to be those stated by the IOM as average or median daily intakes when setting its AI values (Table 3) [38].

Of the 26 products with a Mn salt listed on their labels, 9 (34.6%) contained enough Mn to exceed the IOM ULs when consumed as supplements to ordinary diets. None of the products with no Mn salt listed on their labels would lead to exceedances of the IOM ULs when consumed as supplements to ordinary diets.

We found no difference in cost per day between samples that led to exceedances of the IOM ULs when consumed as supplements and those that did not (W(52) = 179, *p* =.735).

## Limitations

While our sampling methods accounted for products available throughout the US, our survey only examined products available at a single vendor, Amazon.com, during the period December 2021 to January 2022. We may have missed specialty products not available from this vendor. Thus, statistical inferences may be limited to this one vendor rather than to the broader US market, although the vendor we chose has a dominant market share [39]. The ICP-OES detection limit achieved in this study was adequate for the purposes of the study. A lower detection limit could have been achieved if inductively coupled plasma-mass spectrometry (ICP-MS) had been used; however, ICP-MS analyses were not possible due to resource constraints [40]. A further limitation of this study is that, while the IOM DRI values are the closest that the United States has to national standards specifying adequate and tolerable upper intakes of nutrients, including Mn, the reliability of the IOM DRIs is determined by the original studies from which they were derived and the methodology used by the IOM to derive its DRI values. In particular, the IOM DRI methodology may not adequately account for individual variability in dietary or water intakes. Notably, the IOM tolerable UL values we used as the basis for our comparison of exposure estimates have extremely weak scientific support. More recent research on Mn exposure in laboratory animals suggests that the ULs may need to be revised downwards, in which case more of the products we examined here might exceed revised ULs when used as labeled [67]. Additionally, the bioavailability of Mn in supplements has not been well investigated, especially in young children, so it is not known how Mn intakes from supplements may affect Mn exposures or biomarkers [68].

## Conclusions

In this study, we measured manganese levels using ICP-OES in various children's multivitamin and multimineral supplements, including liquids, capsules, tablets, and gummies. Our method detection limit was 0.001 mg/g, with a precision of 0.007 mg/g.

This survey of 52 children's multivitamin and multimineral supplements available on the US market found that most of the products we tested contain more Mn than reported on their labels. Of the 26 products in our survey that contained Mn on their labels, 23 (88.5%) contained more Mn than their labeled value, with an average percent difference of 42.5% (Table 2). Of the 26 products without supplemental Mn listed on their labels, 13 (50.0%) contained detectable concentrations of this element (Table 2).

Of the 26 products we tested without supplemental Mn listed on their labels, none would lead to exceedances of the IOM ULs if consumed in the amounts indicated on the label. In contrast, of the 26 products that we tested with a Mn salt listed on their labels, 4 (15.4%) would exceed the IOM ULs based on their labeled Mn content if consumed with any amount of food or water containing measurable amounts of manganese (Table 4). Of these same 26 products with Mn on their labels, 5 (19.2%) would exceed the IOM ULs based on their measured Mn content, even without additional dietary Mn intake (Table 5). When consumed as supplements to a normal diet, as many as 35% (9/26) of the products that we tested with Mn reported on their labels would lead to exceedances of IOM ULs (Table 6). The cost of the supplements was not correlated with the Mn content, the percent difference between labeled and measured Mn content, or the likelihood of exceeding the IOM ULs.

**Table 6. Total dietary exposure estimates of manganese (Mn) in milligrams per day (mg/day) for children at different ages from 6 months to 5 years, consuming mean (7-12 months old) or median (1-8 years old) dietary intakes of Mn, supplemented with the products containing measurable Mn content.**

| Sample | Total Daily Dietary Intakes of Mn (Mn) Based on Ordinary Diets and Measured Mn Content* | | | | | |
|---|---|---|---|---|---|---|
| | 6 Months | 1 Year | 2 Years | 3 Years | 4 Years | 5 Years |
| | (mg/day) | (mg/day) | (mg/day) | (mg/day) | (mg/day) | (mg/day) |
| **1** | 0.6 | 1.22 | **4.04** | **4.04** | **4.30** | **4.30** |
| 2 | 0.6 | 1.22 | 1.22 | 1.22 | 2.12 | 2.12 |
| 3 | 0.6 | 1.66 | 1.66 | 1.66 | 2.36 | 2.36 |
| **4** | 0.6 | 1.22 | **3.47** | **3.47** | **3.73** | **3.73** |
| **5** | 0.6 | 1.22 | **2.70** | **2.70** | **4.43** | **4.43** |
| **6** | 0.6 | 1.22 | 1.22 | 1.22 | **3.73** | **3.73** |
| **7** | 0.6 | 1.22 | **2.38** | **2.38** | **3.80** | **3.80** |
| 8 | 0.6 | 1.22 | 1.22 | 1.22 | 2.40 | 2.[ |
| 9 | 0.6 | 1.22 | 1.22 | 1.80 | 2.06 | 2.06 |
| 10 | 0.6 | 1.89 | 1.89 | 1.89 | 2.15 | 2.15 |
| 11 | 0.6 | 1.22 | 1.22 | 1.22 | 1.82 | 1.82 |
| 12 | 0.6 | 1.22 | 1.22 | 1.22 | 2.61 | 2.61 |
| 13 | 0.6 | 1.22 | 1.22 | 1.22 | 2.76 | 2.76 |
| 14 | 0.6 | 1.22 | 1.22 | 1.22 | 1.64 | 1.64 |
| 15 | 0.6 | 1.22 | 1.24 | 1.24 | 1.51 | 1.51 |
| 16 | 0.6 | 1.22 | 1.22 | 1.24 | 1.51 | 1.51 |
| **17** | **1.53** | **3.07** | **3.07** | **3.07** | **4.26** | **4.26** |
| 18 | 0.6 | 1.34 | 1.34 | 1.34 | 1.60 | 1.60 |
| 19 | 0.6 | 1.22 | 1.83 | 1.83 | 2.09 | 2.09 |
| **20** | 0.6 | 1.22 | 1.22 | 1.22 | **4.58** | **4.58** |
| 21 | 0.6 | 1.22 | 1.22 | 1.22 | 2.43 | 2.43 |
| 22 | 0.6 | 1.22 | 1.42 | 1.42 | 1.68 | 1.68 |
| **23** | 0.6 | 1.22 | **3.49** | **3.49** | **3.75** | **3.75** |
| **24** | 0.6 | 1.22 | 1.22 | 1.22 | **3.40** | **3.40** |
| 25 | 0.6 | 1.22 | 1.84 | 1.84 | 2.72 | 2.72 |
| 26 | 0.6 | 1.22 | 1.22 | 1.22 | 1.48 | 1.96 |
| 27 | 0.6 | 1.22 | 1.24 | 1.24 | 1.52 | 1.52 |
| 30 | 0.6 | 1.22 | 1.22 | 1.22 | 1.48 | 1.48 |
| 31 | 0.6 | 1.22 | 1.22 | 1.22 | 1.48 | 1.48 |
| 33 | 0.6 | 1.22 | 1.54 | 1.54 | 2.13 | 2.13 |
| 35 | 0.6 | 1.22 | 1.23 | 1.23 | 1.50 | 1.50 |
| 37 | 0.6 | 1.22 | 1.22 | 1.22 | 1.48 | 1.48 |
| 38 | 0.6 | 1.22 | 1.22 | 1.22 | 1.48 | 1.48 |
| 41 | 0.6 | 1.22 | 1.22 | 1.22 | 1.51 | 1.51 |
| 43 | 0.6 | 1.22 | 1.22 | 1.22 | 1.48 | 1.48 |
| 44 | 0.6 | 1.22 | 1.22 | 1.22 | 1.48 | 1.48 |
| 47 | 0.6 | 1.22 | 1.23 | 1.23 | 1.50 | 1.50 |
| 48 | 0.6 | 1.22 | 1.22 | 1.22 | 1.48 | 1.48 |
| 51 | 0.6 | 1.22 | 1.22 | 1.22 | 1.48 | 1.48 |

* Products that exceed the IOM ULs for corresponding age classes when consumed as supplements in addition to ordinary diets according to measured Mn content are highlighted in **bold**. Samples whose measured Mn content was below the detection limit are omitted.

It must be noted that the IOM ULs used for comparison were derived from the maximum intakes of approximately 2 adult individuals in a study that measured no health outcomes [60], so it is possible that the ULs used in the comparisons above are not sufficiently protective for children.

While high dietary intakes of Mn have not been correlated with high blood or serum concentrations of Mn in adults [62], supplemental Mn has been shown to increase blood or serum Mn levels [63–65], and high blood and serum Mn levels have been associated with neurodevelopmental and other adverse health outcomes for children [2,20,21]. The IOM AIs for Mn actually represent the average or median of the observed intakes, since the actual adequate intake levels for this nutrient have never been determined. Thus, it is quite uncertain whether Mn supplements in children are ever beneficial.

To avoid excess exposure to Mn, parents may be advised to choose multivitamin supplements that do not list added Mn on the label, if their pediatricians recommend using multivitamin supplements at all. Manufacturers may be advised to remove Mn from their multivitamin supplements in order to reduce the potential for harm through exceedances of IOM ULs for Mn, or at least reduce the Mn content of their supplements so that the IOM ULs will not be exceeded when the products are consumed alongside ordinary diets.

## Supporting information

**S1 Table. Top 100 best-selling multivitamin and mineral supplements for children listed on Amazon.com (9 September 2025).**
(DOCX)

## Acknowledgments

We acknowledge Norwich University and the University of Vermont for providing institutional support for this project.

## Author contributions

**Conceptualization:** Seth H. Frisbie.

**Data curation:** Seth H. Frisbie.

**Formal analysis:** Erika J. Mitchell.

**Investigation:** Seth H. Frisbie, Amy Hoeltge, Lindsey A. Pett, Molly G. Hoeltge.

**Methodology:** Seth H. Frisbie, Erika J. Mitchell, Amy Hoeltge, Lindsey A. Pett.

**Validation:** Seth H. Frisbie.

**Writing – original draft:** Seth H. Frisbie, Erika J. Mitchell, Amy Hoeltge, Lindsey A. Pett.

**Writing – review & editing:** Seth H. Frisbie, Erika J. Mitchell, Amy Hoeltge, Lindsey A. Pett.

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
