## [Decision Letter · Decision Letter 0]

7 Jan 2026

PONE-D-25-65913Excessive manganese content in children’s multivitamin supplements: potential for neurodevelopmental harm and other adverse health outcomesPLOS One

Dear Dr. Frisbie,

Thank you for submitting your manuscript to PLOS ONE. After careful consideration, we feel that it has merit but does not fully meet PLOS ONE’s publication criteria as it currently stands. Therefore, we invite you to submit a revised version of the manuscript that addresses the points raised during the review process.

We look forward to receiving your revised manuscript.

Kind regards,

Timothy Omara

Academic Editor

PLOS One

Journal Requirements:

The authors have declared that no competing interests exist. EJM’s affiliation is with Better Life Laboratories, Inc., a nonprofit organization that conducts scientific research and provides technical expertise, equipment, and training to help needy people around the world. Better Life Laboratories received no specific funding for this project from any donors. Donors to Better Life Laboratories provided no input in choosing the subject matter of this project, the hypotheses that were tested, the method of analysis, the research findings, or the manner of disseminating the results. This does not alter our adherence to PLOS ONE policies on sharing data and materials.

Additional Editor Comments:

Dear Authors,

The reviewers have recommended reconsideration of the manuscript following major revisions. Please consider the following points during revision:

The use of complementary analytical techniques is generally desirable. However, for trace metal determination, ICP-OES provides greater precision, making the additional use of spectrometric methods, as presented in the Results section, less justifiable. Unless scientifically necessary, ICP-OES should be used as the sole analytical technique in the present study, consistent with what is summarized in the Abstract.Reference articles suggested by reviewers should be cited only if they add value to or improve the manuscript. Editorial decisions are not based on whether such articles are cited.Additional comments and suggestions are provided in the attached MS Word file.

Reviewers' comments:

Reviewer's Responses to Questions

**Comments to the Author**

1. Is the manuscript technically sound, and do the data support the conclusions?

Reviewer #1: Partly

Reviewer #2: No

2. Has the statistical analysis been performed appropriately and rigorously? 

Reviewer #1: Yes

Reviewer #2: No

3. Have the authors made all data underlying the findings in their manuscript fully available?

Reviewer #1: Yes

Reviewer #2: Yes

4. Is the manuscript presented in an intelligible fashion and written in standard English?

Reviewer #1: Yes

Reviewer #2: Yes

5. Review Comments to the Author

Reviewer #1: Dear editor and authors,

The manuscript entitled "Excessive manganese content in children’s multivitamin supplements: potential for neurodevelopmental harm and other adverse health outcomes" evaluated Mn contents in multivitamin and mineral supplements using ICP OES. It presents scientific relevance for Chemistry, Health, Food, Medicine, Biological and others area. The language (English) is satisfactory (but, I suggest the final revision)! However, you need to change some details/information in the Title, Abstract, Introduction, Material and Methods, results and discussion, and conclusions.

1. Title: Adequate!

2. Abstract:

- The abstract is well-written but lacks the main objective of the study. Additionally, it lacks details on the methods and parameters used. What are the optimal conditions for acid digestion? I suggest including information about analytical validation. Has the method been validated? What were the validated parameters? I suggest inserting numerical data related to analytical validation and analyte concentration (range, content in mg/kg).

- Please replace "ICP-OES" with "ICP OES" (without the hyphen) here and throughout the manuscript, as it is not a hyphenated analytical technique.

- Lastly, I suggest highlighting the advantages, innovations, and disadvantages of the proposed method.

- Graphical Abstract/Highlights: ??

3. Introduction section:

- Since an article proposal is also the development/validation of analytical methods, I suggest including more discussions on sample preparation and information/references on analytical techniques (ICP OES) and/or other techniques based on ICP. I suggest consulting and inserting these manuscripts that discuss acid digestion, bioacessibility and ICP technique:

- Total and Bioaccessible contents of Microelements in Multivitamin Formulations Exposed to Consumption in Brazil using in vitro Gastrointestinal Digestion (INFOGEST Protocol). Journal of Trace Elements and Minerals (2025). https://doi.org/10.1016/j.jtemin.2025.100224

- In vitro Monitoring of Macro and Microelements in Multimineral Preparations across Dissolution Profiles by Inductively Coupled Plasma Optical Emission Spectrometry (ICP OES). Journal of the Brazilian Chemical Society (2017). (http://dx.doi.org/10.21577/0103-5053.20170065).

- Analytical approach of elemental impurities in pharmaceutical products: A worldwide review. Spectrochimica Acta Part B-Atomic Spectroscopy (2023). (https://doi.org/10.1016/j.sab.2023.106689)

- Multielement Determination of Macro and Micro Contents in Medicinal Plants and Phytomedicines from Brazil by ICP OES. Journal of the Brazilian Chemical Society (2017). (http://dx.doi.org/10.5935/0103-5053.20160187).

- At the end of the introduction, I suggest at the end of the introduction, to highlight the "innovative" proposal of the method, as well as the advantages/innovation/disadvantages.

4. Material and methods

- In section “Sample Selection”: What are the conditions for collection/acquisition and storage of samples? What is the time/period (from acquisition to analysis)? The text “To assess the accuracy of our results, we also purchased Multielement Tablets Standard Reference Material® (SRM 3294) from the National Institute of Standards and Technology (NIST)” should be moved to section “Quality Assurance and Quality Control”.

- In section “Sample Preparation”: For pretreatment, did the authors follow their own protocol? Or did they follow a reference? If so, please indicate the reference! What is the residual acidity of the samples? Was the dissolved carbon content measured in the samples?

- In section “Sample Analysis by Spectrophotometry”: This information is not included in the abstract! What is the purpose of using molecular absorption spectrophotometry if the proposal is ICP OES? They are different techniques! Was there analytical validation? Did the authors follow their own protocol? Or did they follow a reference?

- In section “Sample Analysis by Inductively Coupled Plasma-Optical Emission Spectroscopy”: How were the ICP OES conditions optimized? Spectral lines? Flux? Did you follow the manufacturers' recommendations or study (uni/multivariately) until the optimal conditions were defined? Has any spectral interference or matrix effect study been carried out?

- In section “Quality Assurance and Quality Control”: And the aspects associated with analytical validation? Have the methods been validated? What protocol was followed? How were the parameters evaluated: LOD, LOQ, precision, accuracy, robustness and linearity? Which concentration ranges were studied? What concentration levels are used to assess accuracy? I suggest detailing the proposed method in more detail... Information on “potential interferents” should be described here!

5. Results and Discussion

The results are interesting, but I suggest expanding the discussions and compare them with other studies published in the literature.

- In section “Accuracy and Precision of the Determination of Manganese Results by Spectrophotometry”: There is no discussion here! I suggest combining it with section “Hypothesis About the Bioavailability of Manganese in Some of the Children’s Supplement Products in our Study” and expanding the discussions and compare them with other studies published in the literature.

- In section “The Detection Limit, Accuracy, and Precision of the Determination of Manganese by Inductively Coupled Plasma-Optical Emission Spectroscopy”: I suggest naming this section: “analytical validation” or “Quality Assurance and Quality Control”. How was the matrix effect evaluated? I suggest expanding the discussions. Please replace "ICP-OES" with "ICP OES" (without the hyphen) here and throughout the manuscript, as it is not a hyphenated analytical technique.

- In section “The Manganese Concentrations Reported on Labels Compared to the Concentrations Measured in Samples”: Table 2 should be better presented, with upper and lower lines. Standard deviation values should be added. There is no discussion here! I suggest incorporating toxicological aspects of Mn into the mansucript for humans. I suggest expanding the discussions and compare them with other studies published in the literature (using the same matrix or others). A good suggestion would be to create a table with data on Mn determination in this matrix in different countries, comparing the sample digestion preparation and analytical technique used.

- In section “Regulations Applicable to Manganese in Children’s Supplements in the United States”: This section should be combined with the previous one “The Manganese Concentrations Reported on Labels Compared to the Concentrations Measured in Samples”; “The Generally Recognized as Safe List and the Labeled Supplemental Manganese Ingredients” and “Governmental Recommendations Applicable to the Manganese Content in Children’s Supplements”. Please, to expand the discussions.

- The sections “Daily Exposure Estimates Based on Labeled Values”, “Daily Exposure Estimates Based on Measured Values” and “Daily Exposure Estimates Based on Ordinary Dietary Intakes in Addition to Measured Values in Supplements” should be combined.

- I suggest, at the end of the "results and discussion", to write a paragraph summarizing the findings and their impacts on the research proposal.

6. Conclusion: I suggest inserting the main findings (numerical data), and to indicate disadvantages/limitations of the method and the study!

7. Table and Figures: Adequate! See comments above.

8. References: I suggest inserting the indicated references into the manuscript. Please, check if the references are in accordance with the journal's rules.

Reviewer #2: Introduction

The Introduction addresses an important and timely issue of potential manganese exposure in children from dietary supplements; however, the argumentation relies primarily on cross-sectional studies and biomarker data whose methodological limitations are not sufficiently discussed. The authors present associations between manganese levels and health effects as largely established, despite the fact that the available evidence is observational in nature and does not demonstrate causality. There are no direct data linking manganese supplementation to actual health disorders in children. The narrative is distinctly alarmist and does not precisely refer to doses that cause documented clinical effects. The bioavailability of manganese from supplements and its interactions with other dietary components are also not discussed, which limits a reliable risk assessment. The formulated research hypotheses are weakly connected to the presented risk narrative, as they are limited to assumptions regarding the agreement between measured and labeled manganese content and non-exceedance of tolerable levels, without directly addressing children’s health or the toxicological mechanisms that are strongly emphasized earlier.

Materials and Methods

The Materials and Methods section describes the analytical procedures in relatively sufficient detail to allow reproduction of the laboratory work; however, it contains significant limitations. A major concern is the sampling strategy. The authors limited product selection exclusively to the Amazon platform, which does not constitute a random or representative sample of the U.S. children’s supplement market. There is no justification as to why Amazon would reflect the actual market structure, nor is there any information on market share of individual brands or brick-and-mortar retail sales (pharmacies, retail chains). The statement that the selected 26 products constitute a “reasonable representation” of the market is not supported by any data and is arbitrary. Consequently, generalization of the results to the entire U.S. market is methodologically unjustified.

There is also a problem with the comparison group. Products without declared manganese were randomly selected from Amazon brands, but their pharmaceutical forms (mainly gummies) differ substantially from those of manganese-containing supplements. There is no justification as to whether and how this control group constitutes an appropriate reference.

An important methodological flaw appears in the statistical section. The calculation of “percent differences” is unusual and non-intuitive (division by the mean of the differences rather than by the labeled value), which may lead to under- or overestimation of relative deviations and hinders interpretation in the context of label compliance. In addition, there is no justification for the selection of statistical tests with respect to specific research hypotheses, and the description of the analyses is very general.

Results and Discussion

This section provides valuable analytical data on manganese content in children’s supplements; however, conclusions regarding bioavailability, health risk, and “exceedances” of ULs are largely over-interpreted, based on simplified assumptions, and lead to a narrative of potential hazard that is not fully justified by the presented data.

In the initial part, the authors demonstrate that the spectrophotometric PAN method is unreliable in the studied matrix, which should be acknowledged as appropriate. However, the subsequent explanation for the method’s failure is problematic. The authors suggest that poor recoveries may indicate strong binding of manganese in supplements and potential “removal” of manganese from the child’s body, which is a far-reaching and unjustified speculation. The digestion conditions (nitric acid + hydrogen peroxide, 120°C) are far more aggressive than physiological conditions, and lack of recovery in such a matrix is a strictly analytical phenomenon and cannot be used to infer bioavailability or possible “sequestration” of minerals in the gastrointestinal tract.

The most serious concern relates to the estimation of exposure and UL exceedances. The authors treat Adequate Intake as equivalent to median intake and use it as an “average dietary background,” which is a simplification and may lead to overestimation of total exposures. Moreover, ULs for manganese in children are based on very limited data, as the authors themselves acknowledge, yet they are nevertheless used as rigid toxicity thresholds, creating an alarmist narrative of “exceedances” without accounting for the uncertainty of these values or individual variability in dietary and water-borne manganese intake. The regulatory discussion and references to the GRAS list also contain significant inaccuracies. The absence of a particular manganese form from the GRAS list does not automatically imply illegality or unsafety in dietary supplements, which is indirectly suggested and may mislead readers.

Conclusions

The conclusions presented in the Conclusions section largely go beyond what can be directly inferred from the empirical data obtained in the study, and the narrative has a distinctly normative and alarmist character that is not fully justified by the scope and quality of the collected data.

6. PLOS authors have the option to publish the peer review history of their article (what does this mean? ). If published, this will include your full peer review and any attached files.

**Do you want your identity to be public for this peer review?** For information about this choice, including consent withdrawal, please see our Privacy Policy .

Reviewer #1: No

Reviewer #2: No

---

## [Author Response · Author response to Decision Letter 1]

5 Feb 2026

PONE-D-25-65913

Excessive manganese content in children’s multivitamin supplements: potential for neurodevelopmental harm and other adverse health outcomes

Journal Requirements:

Response: We have adjusted the formatting to meet the journal’s style requirements.

The authors have declared that no competing interests exist. EJM’s affiliation is with Better Life Laboratories, Inc., a nonprofit organization that conducts scientific research and provides technical expertise, equipment, and training to help needy people around the world. Better Life Laboratories received no specific funding for this project from any donors. Donors to Better Life Laboratories provided no input in choosing the subject matter of this project, the hypotheses that were tested, the method of analysis, the research findings, or the manner of disseminating the results. This does not alter our adherence to PLOS ONE policies on sharing data and materials.

Response: The required sentence about adhering to PLOS ONE policies is already included at the end of our original Competing Interests section. Therefore, no changes were made.

Response: We have added captions for the supporting information files and made the in-text citations consistent.

Response: We have read and evaluated the recommended references. We have used and cited one of these references.

Additional Editor Comments:

Dear Authors,

The reviewers have recommended reconsideration of the manuscript following major revisions. Please consider the following points during revision:

1.

The use of complementary analytical techniques is generally desirable. However, for trace metal determination, ICP-OES provides greater precision, making the additional use of spectrometric methods, as presented in the Results section, less justifiable. Unless scientifically necessary, ICP-OES should be used as the sole analytical technique in the present study, consistent with what is summarized in the Abstract.

Response: We have deleted the spectrophotometry sections as suggested. Accordingly, we have deleted Table 1, references 40 and 44, and Supplementary Table S2.

2.

Reference articles suggested by reviewers should be cited only if they add value to or improve the manuscript. Editorial decisions are not based on whether such articles are cited.

Response: We have read and evaluated the recommended references. We have used and cited one of these references.

3.

Additional comments and suggestions are provided in the attached MS Word file.

Response: We have addressed all comments and suggestions in this MS Word file.

Reviewers' comments:

Reviewer's Responses to Questions

Comments to the Author

1. Is the manuscript technically sound, and do the data support the conclusions?

Reviewer #1: Partly

Reviewer #2: No

2. Has the statistical analysis been performed appropriately and rigorously?

Reviewer #1: Yes

Reviewer #2: No

3. Have the authors made all data underlying the findings in their manuscript fully available?

Reviewer #1: Yes

Reviewer #2: Yes

4. Is the manuscript presented in an intelligible fashion and written in standard English?

Reviewer #1: Yes

Reviewer #2: Yes

5. Review Comments to the Author

Reviewer #1: Dear editor and authors,

The manuscript entitled "Excessive manganese content in children’s multivitamin supplements: potential for neurodevelopmental harm and other adverse health outcomes" evaluated Mn contents in multivitamin and mineral supplements using ICP OES. It presents scientific relevance for Chemistry, Health, Food, Medicine, Biological and others area. The language (English) is satisfactory (but, I suggest the final revision)! However, you need to change some details/information in the Title, Abstract, Introduction, Material and Methods, results and discussion, and conclusions.

1. Title: Adequate!

2. Abstract:

- The abstract is well-written but lacks the main objective of the study.

Response: We added the following sentence to the end of the first paragraph of the abstract to make the main objective more clear: In this study, we measured Mn concentrations in children’s multivitamin and mineral supplements and compared the concentrations to the Institute of Medicine’s (IOM’s) Upper tolerable Levels (ULs).

Additionally, it lacks details on the methods and parameters used.

Response: We were unable to add details on the methods and parameters used to the abstract without exceeding the 300-word limit for the abstract. The details are, however, included in the “Materials and methods” section.

What are the optimal conditions for acid digestion?

Response: We were unable to add details on the conditions used for acid digestion without exceeding the 300-word limit for the abstract. The details are, however, included in the “Materials and methods” section.

I suggest including information about analytical validation. Has the method been validated? What were the validated parameters? I suggest inserting numerical data related to analytical validation and analyte concentration (range, content in mg/kg).

Response: We were unable to add details on the analytical validation to the abstract without exceeding the 300-word limit for the abstract. The details are, however, included in the “Results and discussion” section.

- Please replace "ICP-OES" with "ICP OES" (without the hyphen) here and throughout the manuscript, as it is not a hyphenated analytical technique.

Response: Actually, ICP-OES is considered a “hyphenated analytical technique”; see, for instance, https://www.unine.ch/npac/wp-content/uploads/sites/140/ICPOES_Powerful-analytical-technique.pdf. The ICP-OES technique combines 2 different analytical instruments to leverage their individual strengths, which is the definition of a “hyphenated method.” It is commonly abbreviated as “ICP-OES” by the major manufacturers of ICP-OES equipment, including Agilent (https://www.agilent.com/en/support/atomic-spectroscopy/inductively-coupled-plasma-optical-emission-spectroscopy-icp-oes/icp-oes-faq) and ThermoFisher Scientific (https://www.thermofisher.com/mq/en/home/industrial/spectroscopy-elemental-isotope-analysis/spectroscopy-elemental-isotope-analysis-learning-center/trace-elemental-analysis-tea-information/icp-oes-information.html), as well as governmental organizations such as the United States Geological Survey (https://www.usgs.gov/labs/national-water-quality-laboratory/science/science-topics/inductively-coupled-plasma-optical). Thus, we have retained the commonly used and accepted abbreviation “ICP-OES”.

- Lastly, I suggest highlighting the advantages, innovations, and disadvantages of the proposed method.

Response: We were unable to add details about the advantages and disadvantages of the method to the abstract without exceeding the 300-word limit for the abstract. We included a brief discussion comparing the advantages and disadvantages of ICP-OES and ICP-MS in the new “Limitations” section at the end of the Results and discussion section.

- Graphical Abstract/Highlights: ??

Response: To our knowledge, PLoSOne does not make use of graphical abstracts or highlights; please see https://storage.googleapis.com/plos-published-prod/wjVg/PLOSOne_formatting_sample_main_body.pdf

3. Introduction section:

- Since an article proposal is also the development/validation of analytical methods, I suggest including more discussions on sample preparation and information/references on analytical techniques (ICP OES) and/or other techniques based on ICP.

Response: We added text at the end of the introduction to give a concise overview of our analytical techniques.

I suggest consulting and inserting these manuscripts that discuss acid digestion, bioacessibility and ICP technique:

- Total and Bioaccessible contents of Microelements in Multivitamin Formulations Exposed to Consumption in Brazil using in vitro Gastrointestinal Digestion (INFOGEST Protocol). Journal of Trace Elements and Minerals (2025). https://doi.org/10.1016/j.jtemin.2025.100224

- In vitro Monitoring of Macro and Microelements in Multimineral Preparations across Dissolution Profiles by Inductively Coupled Plasma Optical Emission Spectrometry (ICP OES). Journal of the Brazilian Chemical Society (2017). (http://dx.doi.org/10.21577/0103-5053.20170065).

- Analytical approach of elemental impurities in pharmaceutical products: A worldwide review. Spectrochimica Acta Part B-Atomic Spectroscopy (2023). (https://doi.org/10.1016/j.sab.2023.106689)

- Multielement Determination of Macro and Micro Contents in Medicinal Plants and Phytomedicines from Brazil by ICP OES. Journal of the Brazilian Chemical Society (2017). (http://dx.doi.org/10.5935/0103-5053.20160187).

Response: We appreciate the suggested references. We have examined and discussed them. We have added a citation to the 2025 reference listed above to our “Results and discussion” section.

- At the end of the introduction, I suggest at the end of the introduction, to highlight the "innovative" proposal of the method, as well as the advantages/innovation/disadvantages.

Response: Our analytical method, ICP-OES, is simple and straightforward. Our overall methodology follows the standard methodology for analysis in similar matrices described in the instrument manual, so we do not consider it innovative. Our quality assurance procedures demonstrated that it was quite adequate for our main objective of measuring Mn in supplements. Since we do not consider this method innovative, we did not make any further changes to the “Introduction” section.

4. Material and methods

- In section “Sample Selection”: What are the conditions for collection/acquisition and storage of samples? What is the time/period (from acquisition to analysis)?

Response: We have added information about acquisition date and storage of samples to the Sample selection section.

The text “To assess the accuracy of our results, we also purchased Multielement Tablets Standard Reference Material® (SRM 3294) from the National Institute of Standards and Technology (NIST)” should be moved to section “Quality Assurance and Quality Control”.

Response: The first sentence of the “Sample preparation” section refers to the NIST reference materials, so the purchase of the materials must be explained before this sentence. That is why we included the mention of the purchase of the NIST reference materials at the end of the “Sample selection” section. No changes made.

- In section “Sample Preparation”: For pretreatment, did the authors follow their own protocol? Or did they follow a reference? If so, please indicate the reference! What is the residual acidity of the samples? Was the dissolved carbon content measured in the samples?

Response: Thank you for your comment. We followed the operational instructions provided in the ICP-OES manufacturer’s manual. We added a citation to this manual in the “Materials and methods” section. We did not measure residual acidity or dissolved carbon. More importantly, we measured the percent recovery of manganese from the National Institute of Standards and Technology (NIST) Multielement Tablets Standard Reference Material® (SRM 3294) in triplicate. As we reported in our paper, all three of these measurements of the SRM, 1.36 mg/g, 1.43 mg/g, and 1.44 mg/g, were within the acceptable range of 1.33 mg/g to 1.55 mg/g specified by the NIST. Therefore, we conclude, based on this internationally recognized standard, that our measured manganese concentrations are accurate.

- In section “Sample Analysis by Spectrophotometry”: This information is not included in the abstract! What is the purpose of using molecular absorption spectrophotometry if the proposal is ICP OES? They are different techniques! Was there analytical validation? Did the authors follow their own protocol? Or did they follow a reference?

Response: In response to suggestions by the editor, we have deleted the section on spectrophotometry.

- In section “Sample Analysis by Inductively Coupled Plasma-Optical Emission Spectroscopy”: How were the ICP OES conditions optimized? Spectral lines? Flux? Did you follow the manufacturers' recommendations or study (uni/multivariately) until the optimal conditions were defined? Has any spectral interference or matrix effect study been carried out?

Response: We followed the operational instructions provided in the ICP-OES manufacturer’s manual. For clarity, we added text to the Materials and methods section with additional details about instrument parameters. With respect to your question, “Has any spectral interference or matrix effect study been carried out?” Yes, we did evaluate these potential matrix effects. As we reported in our paper, “The percent recoveries of known additions of standard to samples were measured at 92.7%, 99.0%, 102%, and 112% by inductively coupled plasma-optical emission spectroscopy. The 95% confidenc

---

## [Decision Letter · Decision Letter 1]

9 Feb 2026

Excessive manganese content in children’s multivitamin supplements: potential for neurodevelopmental harm and other adverse health outcomes

PONE-D-25-65913R1

Dear Dr. Frisbie,

We’re pleased to inform you that your manuscript has been judged scientifically suitable for publication and will be formally accepted for publication once it meets all outstanding technical requirements.

Kind regards,

Timothy Omara

Academic Editor

PLOS One

Additional Editor Comments (optional):

Reviewers' comments:

Reviewer's Responses to Questions

**Comments to the Author**

1. If the authors have adequately addressed your comments raised in a previous round of review and you feel that this manuscript is now acceptable for publication, you may indicate that here to bypass the “Comments to the Author” section, enter your conflict of interest statement in the “Confidential to Editor” section, and submit your "Accept" recommendation.

Reviewer #1: All comments have been addressed

Reviewer #2: All comments have been addressed

2. Is the manuscript technically sound, and do the data support the conclusions?

Reviewer #1: Yes

Reviewer #2: Yes

3. Has the statistical analysis been performed appropriately and rigorously? 

Reviewer #1: N/A

Reviewer #2: Yes

4. Have the authors made all data underlying the findings in their manuscript fully available?

Reviewer #1: Yes

Reviewer #2: Yes

5. Is the manuscript presented in an intelligible fashion and written in standard English?

Reviewer #1: Yes

Reviewer #2: Yes

6. Review Comments to the Author

Reviewer #1: Although they did not agree with some of my suggestions, the authors improved the manuscript. Accept.

Reviewer #2: Thank you to the authors for their appropriate adjustments to the sections of concern and their astute observations and comments back.

7. PLOS authors have the option to publish the peer review history of their article (what does this mean? ). If published, this will include your full peer review and any attached files.

**Do you want your identity to be public for this peer review?** For information about this choice, including consent withdrawal, please see our Privacy Policy .

Reviewer #1: No

Reviewer #2: No

---

## [Editor Report · Acceptance letter]

PONE-D-25-65913R1

PLOS One

Dear Dr. Frisbie,

I'm pleased to inform you that your manuscript has been deemed suitable for publication in PLOS One. Congratulations! Your manuscript is now being handed over to our production team.

Kind regards,

on behalf of

Dr. Timothy Omara

Academic Editor

PLOS One